# TOWARDS SPATIO-TEMPORALLY CONSISTENT 4D RECONSTRUCTION FROM SPARSE CAMERAS

## ABSTRACT

High-quality 4D reconstruction enables photorealistic and immersive rendering of the dynamic real world. However, unlike static scenes that can be fully captured with a single camera, high-quality dynamic scene benchmarks typically use dense arrays of approximately 20 synchronized cameras or more. The reliance on such costly lab setups severely limits practical scalability. To this end, we propose a sparse-camera dynamic reconstruction framework that exploits abundant yet inconsistent generative observations. Our key innovation is the Spatio-Temporal Distortion Field, which provides a unified mechanism for modeling inconsistencies in generative observations across both spatial and temporal dimensions. Building on this, we develop a complete pipeline that enables 4D reconstruction from sparse and uncalibrated camera inputs. We evaluate our method on multi-camera dynamic scene benchmarks, achieving spatio-temporally consistent high-fidelity renderings and significantly outperforming existing approaches.

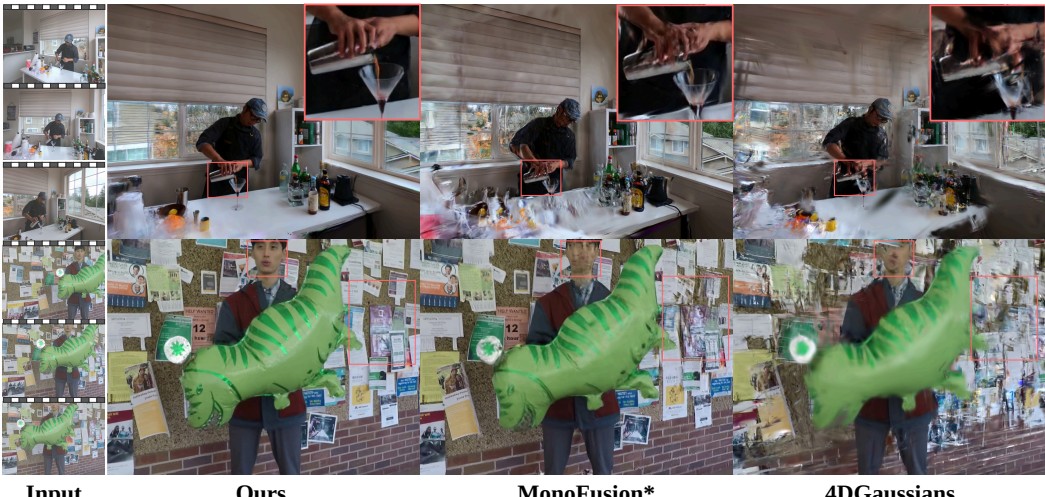

| **Input** | **Ours** | **MonoFusion*** | **4DGaussians** |

Figure 1: **Novel view rendering comparison.** Our method enables high-quality, cost-effective dynamic scene reconstruction using only 2-3 cameras, producing spatio-temporally consistent and photorealistic results. Please refer to our **supplementary video** for further dynamic comparisons.

## 1 INTRODUCTION

Advances in dynamic scene novel view synthesis (NVS), particularly real-time 4D Gaussian Splatting (4DGS), have enabled high-fidelity dynamic rendering and hold great potential for applications in VR/AR, film production, short videos, live streaming, etc. (Newcombe et al., 2015).

However, immersive 4D reconstruction generally requires dense camera inputs. Unlike static scenes that can be captured with a single camera, high-quality dynamic scene benchmarks (Li et al., 2022; Sabater et al., 2017; Yoon et al., 2020) are typically constructed using dense camera arrays com-

prising approximately 20 synchronized cameras. Such costly lab setups severely hinder broader adoption and scalability.

Therefore, we aim to reconstruct high-fidelity dynamic scenes from sparse cameras. Sparse-view 4D reconstruction remains largely unsolved (Younis & Cheng, 2025), as spatio-temporally consistent reconstruction of complex motions relies heavily on dense observations. Sparse inputs aggravate the ill-posed nature of 4D reconstruction (Jin et al., 2025), making this task highly challenging.

Geometric regularization is an intuitive strategy for sparse-view reconstruction (Younis & Cheng, 2025). MonoFusion (Wang et al., 2025b), followed with Shape-of-Motion (Wang et al., 2024a), uses depth and tracking regularization to align 4D scene content and improve multi-view consistency in novel view synthesis. However, these regularization techniques mainly focus on structural constraints and are insufficient to preserve accurate appearance, causing the rendering quality to quickly collapse under viewpoint shifts, as shown in Fig. 1, which prevents free-viewpoint exploration.

With the remarkable progress in Camera-Controlled Video Diffusion Models (Yu et al., 2024; Bai et al., 2025), another intuitive direction is to leverage such models to generate high-quality spatio-temporal data, thereby providing additional observation for 4D reconstruction. However, these photorealistic generated results often exhibit spatio-temporal inconsistencies, such as flickering surfaces and unstable object motions across views and time, as shown in Fig. 2, which undermine the coherence of dynamic scenes and cause severe blurring and artifacts.

To this end, our key innovation is the Spatio-Temporal Distortion Field (STDF), a lightweight mechanism enables unified modeling of inconsistencies in generative observations across both space and time. Notably, the STDF is discarded after training, thus introducing zero additional computational overhead to novel view rendering. Moreover, due to the difficulty of obtaining accurate pose priors from sparse inputs, we conduct experiments using uncalibrated sparse views. Our pipeline jointly optimizes pose, rendering, and smoothness terms to produce spatio-temporally consistent dynamic reconstructions.

Finally, we validate our approach on three standard 4D reconstruction benchmarks, including Neural 3D Video (Li et al., 2022), Technicolor (Sabater et al., 2017), and Nvidia Dynamic Scenes (Yoon et al., 2020). To the best of our knowledge, this is the first work to achieve sparse-camera 4D reconstruction on dynamic scene benchmarks, evaluated across all camera views.

Our contributions are as follows: **(i)** We propose unified modeling of spatio-temporal inconsistencies in generative observations by introducing the Spatio-Temporal Distortion Field (STDF). **(ii)** We present a complete pipeline and optimization strategy that supports high-fidelity 4D scene reconstruction from uncalibrated sparse inputs. **(iii)** Extensive experiments on multi-camera 4D benchmarks show that our method outperforms prior approaches, enabling photorealistic and spatio-temporally consistent novel view rendering of dynamic scenes from sparse-camera inputs.

## 2 RELATED WORK

### 2.1 SPARSE-VIEW DYNAMIC RECONSTRUCTION

Sparse-view dynamic reconstruction is still in its early stages. Some studies have initially focused on reconstructing dynamic content of objects or human bodies using sparse cameras. Works such as Peng et al. (2021b); Weng et al. (2022); Hu et al. (2024); Peng et al. (2021a) build a canonical static 3D space based on SMPL priors and learn a deformation field to map it to Gaussian primitives at different time steps. Jin et al. (2025) utilize existing human datasets and train a spatio-temporal diffusion model under the guidance of SMPL priors to generate additional temporally consistent multi-view human videos for reconstruction. Research on sparse-view dynamic reconstruction in real-world scenes remains limited. A recent line of work explores dynamic reconstruction from monocular videos (Wang et al., 2024a; Lei et al., 2024; Liu et al., 2025). However, these methods largely rely on monocular depth and tracking-based regularization, and without multi-view constraints, the rendering of novel views quickly collapses under viewpoint shifts. MonoFusion (Wang et al., 2025b) improves upon this line by integrating monocular depth, 3D tracking information, and DINOv2 (Oquab et al., 2023) features, extending monocular 4D reconstruction methods to sparse-view settings. Nevertheless, results show that under these geometric regularizations, the rendering

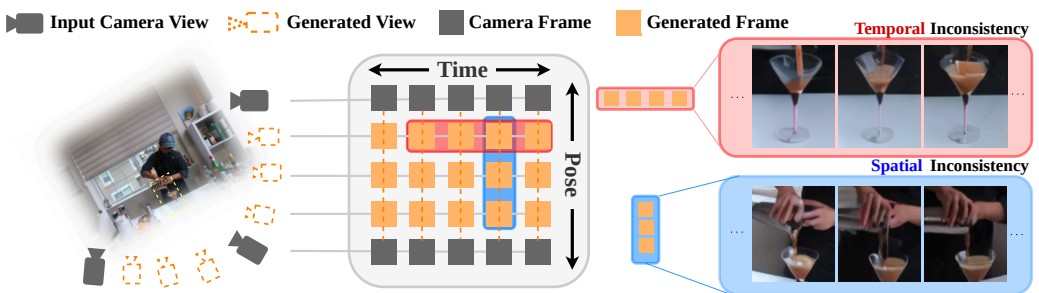

Figure 2: **Spatio-temporal inconsistency.** Real cameras (grey) capture consistent content of multi-view dynamic scene, while generative results (orange) include additional observations at different poses and time. Inconsistencies across poses at the same time are referred to as *spatial inconsistencies*, and inconsistencies across time at the same pose are referred to as *temporal inconsistencies*.

quality remains suboptimal, exhibiting floating artifacts, missing details, and geometric distortions, making it challenging to achieve photorealistic renderings in novel view.

## 2.2 Video Diffusion Model for Novel View Synthesis.

Recent advances in video diffusion models (VDMs) have enabled generating continuous videos from a single image prompt. In the domain of novel view synthesis, several works (Liu et al., 2024; Long et al., 2024; Shi et al., 2023; Wu et al., 2024b; Ye et al., 2024) directly focus on multi-view consistent video generation, but they are mostly constrained to object-centric setting. For scene-level generation, explorations such as Sun et al. (2024); Wu et al. (2025); Wang et al. (2025a) have been attempted, yet their performance in complex realistic environments remains unsatisfactory due to the lack of sufficient training data. Another line of work, camera-controlled video diffusion model, incorporates camera motion control during generation for controllable novel-view videos. Wang et al. (2024b) introduces a Camera Motion Control Module that injects camera extrinsic parameters into temporal transformers. Xu et al. (2024); He et al. (2024); Bai et al. (2025) further improve controllability by replacing explicit camera pose parameters with Plücker ray embeddings. Yu et al. (2024) incorporate point cloud priors to improve motion control stability.

Despite these advances, generated frames still suffer from spatial and cross-view inconsistencies, limiting their applicability to 4D scene reconstruction. To address these challenges, we propose a novel pipeline that explicitly enforces consistency across space, time, and viewpoints.

## 3 Method

In this section, we present a new framework that leverages the capability of video diffusion models to provide auxiliary observations, enabling 4D scene reconstruction from a limited number of input views. Sec 3.2 introduces the overall framework. Sec 3.3 details the core component - Spatio-Temporal Distortion Field - which helps to incorporate generated images into 4D scene reconstruction. Finally, Sec 3.4 describes the optimization scheme.

## 3.1 Preliminary

**4D Gaussian Splatting.** A line of 4DGS approaches directly models Gaussian primitives in 4D space to represent the dynamic scene. In this paradigm, the temporal axis is treated as an additional independent coordinate dimension, such that 3D Gaussians are directly lifted into 4D. Each 4D Gaussian is parameterized by its center position $\boldsymbol{\mu} = (\mu_x, \mu_y, \mu_z, \mu_t)$ and a covariance matrix $\boldsymbol{\Sigma} \in \mathbb{R}^{4\times4}$, where $\boldsymbol{\Sigma}$ is decomposed into a scaling matrix $\boldsymbol{S} = \mathrm{diag}(s_x, s_y, s_z, s_t)$ and a rotation matrix $\boldsymbol{R} \in \mathbb{R}^{4\times4}$. Same as 3D Gaussians, each 4D primitive maintains a set of SH coefficients and an opacity $\alpha$.

To parameterize 4D rotation matrix in Euclidean space, algebraic geometry tools such as pair of isotropic rotations Yang et al. (2023) $\boldsymbol{R} = L(\boldsymbol{q}_l)R(\boldsymbol{q}_r)$ or a normalized 4D rotor $\mathbf{r}$ with 8 coeffi-

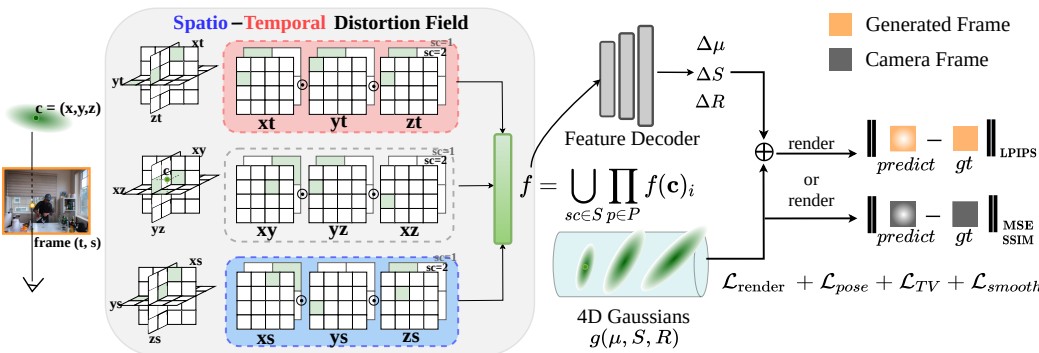

Figure 3: **Method overview.** Given a generated frame at temporal index $t$ and pose index $s$, each 4D Gaussian at $c = (x, y, z)$ is projected onto the planes of the Spatio-Temporal Distortion Field to obtain deformation features, which are then decoded by a small MLP to produce the deformation values. We use separate photometric losses for real and generated frames, and additionally introduce regularization terms on pose, feature plane, and spatial smoothness to enhance optimization stability.

cients Duan et al. (2024) $\boldsymbol{R} = \mathcal{F}_{map}(\mathcal{F}_{norm}(\mathbf{r}))$ are employed, which are mathematically equivalent. At a given timestamp $t$, temporal slicing is performed to project the attributes of each 4D Gaussian into the corresponding 3D subspace as follows:

$$\mathcal{G}_{3D}(\boldsymbol{x}, t) = e^{-\frac{1}{2}\lambda(t-\mu_t)^2} e^{-\frac{1}{2}[\boldsymbol{x}-\boldsymbol{\mu}(t)]^T \boldsymbol{\Sigma}_{3D}^{-1}[\boldsymbol{x}-\boldsymbol{\mu}(t)]}, \tag{1}$$

The rendering process follows the standard differential splatting procedure of 3DGS, while the densification is performed in both spatial and temporal dimensions.

**K-planes Factorization.** K-planes (Fridovich-Keil et al., 2023) introduces a simple and interpretable representation for arbitrary $d$-dimensional scenes, referred to as *K-planes factorization*. In this framework, $k = \binom{d}{2}$ planes are employed to represent every combinations of two dimensions. Taking an $d$-dimensional coorinate as input, K-planes maps it to a feature vector, which is then decoded by a tiny MLP to obtain target attribute value. For dynamic 4D scenes, this results in the so-called *hex-planes*, consisting of three space-only planes $xy$, $xz$, $yz$ and three space-time planes $xt$, $yt$, $zt$. Such a representation has been widely adopted in both deformation-based NeRF and 4DGS methods. NeRFs use it to estimate a world-to-canonical mapping $\mathcal{M} : (\boldsymbol{p}, t) \to \Delta\boldsymbol{p}$, where $\boldsymbol{p}$ reveals to the world spatial point and $t$ is the target time. Then the vanilla NeRF pipeline is applied with canonical spatial point $\boldsymbol{p} + \Delta\boldsymbol{p}$ and view direction $\boldsymbol{d}$ as input. 4DGS using it to compute the canonical-to-world mapping $\mathcal{F} : (\mathcal{G}, t) \to \Delta\mathcal{G}$ for each attribute of a canonical 3D Gaussian primitive $\mathcal{G}$ at time $t$. An image $I$ with view matrix $\mathbf{M} = [\mathbf{R}|\mathbf{T}]$ is rendered by the differential splatting with the deformed 3D Gaussians $\mathcal{G}'$ following $I = \mathcal{S}(\mathbf{M}, \mathcal{G}')$, where $\mathcal{G}' = \mathcal{G} + \Delta\mathcal{G}$.

### 3.2 FRAMEWORK

Given $N$ sparse input camera videos with $L$ frames, our goal is to optimize a 4DGS model using auxiliary generated sequences. For simplicity, we refer to the set of images from $N$ input video sequences and their corresponding camera poses as *input views* $V_I = \{(I_s^t, [\mathbf{R}|\mathbf{T}]_s)|t = 0, ..., L; s = 0, ..., N\}$, and the set of images $I$ from $M$ generated video sequences with their poses $[\mathbf{R}|\mathbf{T}]$ as *generated views* $V_G = \{(I_s^t, [\mathbf{R}|\mathbf{T}]_s)|t = 0, ..., L; s = 0, ..., M\}$. The 4DGS model is trained with $V_I + V_G$.

The noising and denoising process of video diffusion models introduces severe geometric inconsistencies across space and time during generation. If such generated views are directly used for scene reconstruction, these inconsistencies will significantly degrade the geometric consistency of 4DGS, leading to noticeable artifacts. Therefore, when leveraging such diffusion-based observations to assist reconstruction, it is crucial to extract and disentangle these inconsistencies to construct canonical 4D Gaussians.

Our framework consists of 4D Gaussians $\mathcal{G}_{4D}$ and a spatio-temporal distortion field $\mathcal{F}$ that models the inconsistencies in each generated view $V_G^{t,s} \in V_G$, formally represented as:

$$\mathcal{F} : (\mathcal{G}_{4D}, t, s) \to \Delta\mathcal{G}_{4D}. \tag{2}$$

where $t$ denotes the time index, while $s$ denotes the pose index, as illustrated in Fig. 2. Through the proposed distortion field, the variation of canonical 4D Gaussians on a generated view can be obtained, which then yields the distorted 4D Gaussians $\mathcal{G}'_{4D} = \mathcal{G}_{4D} + \Delta\mathcal{G}_{4D}$. Our framework converts the original 4D Gaussians $\mathcal{G}_{4D}$ into another group of 4D Gaussians $\mathcal{G}'_{4D}$ given the index $(t, s)$ of a generated view to model its distortion, with the differential splatting still effective.

## 3.3 SPATIO-TEMPORAL DISTORTION FIELD

Specifically, the spatio-temporal distortion field $\mathcal{F}$ consists of an Ennea-plane representation and a lightweight multi-head MLP serving as the fused feature decoder. We factorize the 5D volume defined by $(x, y, z, t, s)$ into $k = \binom{5}{2} = 10$ two-dimensional planes, each corresponding to a pair of dimensions. Since the combination $(t, s)$ does not encode any form of distortion, this plane is omitted. Consequently, such factorization decomposes the 5D neural voxel into nine multi-resolution 2D feature planes $\boldsymbol{P} = \{\boldsymbol{P}_{xy}, \boldsymbol{P}_{xz}, \boldsymbol{P}_{yz}, \boldsymbol{P}_{xt}, \boldsymbol{P}_{yt}, \boldsymbol{P}_{zt}, \boldsymbol{P}_{xs}, \boldsymbol{P}_{ys}, \boldsymbol{P}_{zs}\}$. Each feature plane is defined as $\boldsymbol{P}_{ij} \in \mathbb{R}^{lN_i \times lN_j \times h}$, where $h$ denotes the feature dimension, $N_i$ and $N_j$ represent the basis resolution of the corresponding two axes, and $l$ is the scale factor for multi-resolution structure.

Given a 5D coordinate $\boldsymbol{c} = \{x, y, z, t, s\}$, the corresponding feature vector is obtained as follows. First, each dimension of $\boldsymbol{c}$ is normalized to its resolution range $[0, N_i)$, and then coordinate $\boldsymbol{c}$ is projected onto the nine planes aforementioned. The feature of $\boldsymbol{c}$ on each plane is extracted via bilinear interpolation, formally:

$$\boldsymbol{f}(\boldsymbol{c})_c = \mathrm{interp}(\boldsymbol{P}_c, \pi_c(\boldsymbol{c})), \quad c \in \{xy, xz, yz, xt, yt, zt, xs, ys, zs\} \tag{3}$$

where $\pi_c$ denotes the projection of $\boldsymbol{c}$ onto the corresponding plane, and 'interp' indicates the bilinear interpolation over the 2D grid. The features extracted from the feature planes are then fused by element-wise multiplication to obtain an $h$-dimensional feature vector at a given resolution. Features across different resolutions are then concatenated to form the final features.

$$\boldsymbol{f}(\boldsymbol{c}) = \bigcup_{sc} \prod_{\boldsymbol{P}_c \in \boldsymbol{P}} \boldsymbol{f}(\boldsymbol{c})_c \tag{4}$$

These features are decoded by a multi-head MLP decoder $\mathcal{D} = \{\phi, \phi_{\boldsymbol{p}}, \phi_{\boldsymbol{q}_l}, \phi_{\boldsymbol{q}_r}, \phi_{\boldsymbol{s}}\}$ into the distortion of various 4D Gaussian attributes, including position $\Delta\boldsymbol{\mu} = \phi_{\boldsymbol{\mu}}(\phi(\boldsymbol{f}))$, rotation $\Delta\boldsymbol{q}_l = \phi_{\boldsymbol{q}_l}(\phi(\boldsymbol{f})), \Delta\boldsymbol{q}_r = \phi_{\boldsymbol{q}_r}(\phi(\boldsymbol{f}))$, and scaling $\Delta\boldsymbol{s} = \phi_{\boldsymbol{s}}(\phi(\boldsymbol{f}))$. Then, the distorted attributes can be computed as:

$$(\boldsymbol{\mu}', \boldsymbol{q}'_l, \boldsymbol{q}'_r, \boldsymbol{s}') = (\boldsymbol{\mu} + \Delta\boldsymbol{\mu}, \boldsymbol{q}_l + \Delta\boldsymbol{q}_l, \boldsymbol{q}_r + \Delta\boldsymbol{q}_r, \boldsymbol{s} + \Delta\boldsymbol{s}) \tag{5}$$

During training, the distorted Gaussians are used to render the generated views, while the original Gaussians are used to render the real views. After training, the distortion terms are discarded, and only the canonical 4D Gaussian is retained.

## 3.4 OPTIMIZATION

**Pose Optimization.** Because of the nconsistencies present in the generated video frames, the alignment accuracy is compromised when using traditional COLMAP (Schönberger & Frahm, 2016; Schönberger et al., 2016) for estimating camera extrinsics. To mitigate this issue, we propose simultaneous optimization of camera extrinsics along with the 4D Gaussian attributes, treating camera extrinsics as learnable variables likewise.

**Loss Function.** For input views, we apply the standard photometric loss. The photometric loss includes an $\mathcal{L}_1$ RGB loss and a D-SSIM loss (Wang et al., 2004).

$$\mathcal{L}_{\mathrm{input}} = (1 - \lambda)\mathcal{L}_1 + \lambda\mathcal{L}_{\text{D-SSIM}}. \tag{6}$$

For generated views, applying standard photometric loss directly leads to degraded reconstruction quality due to the inherent distortions in generated frames. To address this, we propose using perceptual loss (Zhang et al., 2018) to supervise texture and reconstruction similarity.

$$\mathcal{L}_{\mathrm{gen}} = \lambda_1\mathcal{L}_1 + \lambda_2\mathcal{L}_{lpips}. \tag{7}$$

In order to prevent the optimized pose from deviating significantly from their original initialization, we introduce an additional constraint loss:

$$\mathcal{L}_{\text{pose}} = \lambda_p(||\boldsymbol{T} - \hat{\boldsymbol{T}}|| + ||\boldsymbol{q} - \hat{\boldsymbol{q}}||), \tag{8}$$

where $\boldsymbol{T}$ and $\boldsymbol{q}$ represent the optimized translation and rotation of a camera, $\hat{\boldsymbol{T}}$ and $\hat{\boldsymbol{q}}$ are the corresponding initial extrinsics obtained from COLMAP (Schönberger & Frahm, 2016; Schönberger et al., 2016), and parameter $\lambda_p$ balances the camera optimization term with other loss components.

Additionally, following K-Planes (Fridovich-Keil et al., 2023), a grid-based total variation loss $\mathcal{L}_{\text{TV}}$ is also applied for spatial smoothness. Since the distortions in the generated images from Yu et al. (2024) are continuous along the pose axis but exhibit abrupt changes along the time axis, we apply a smoothness regularization over pose axis with a second derivative filter:

$$\mathcal{L}_{\text{smooth}} = \lambda_s \frac{1}{|C|} \sum_{c \in C} \frac{1}{N_i N_s} \sum_{i,s} ||(\boldsymbol{P}_c^{i,s-1} - \boldsymbol{P}_c^{i,s}) - (\boldsymbol{P}_c^{i,s} - \boldsymbol{P}_c^{i,s+1})||_2^2, \quad C = \{xs, ys, zs\}, \tag{9}$$

where $i$, $s$ are indices on plane $\boldsymbol{P}_c$, $N_i$ and $N_s$ represent the resolution of each axis. Overall, the total loss can be formulated as:

$$\mathcal{L} = \mathcal{L}_{\text{input}} + \mathcal{L}_{\text{gen}} + \mathcal{L}_{\text{pose}} + \mathcal{L}_{\text{TV}} + \mathcal{L}_{\text{smooth}} \tag{10}$$

## 4 EXPERIMENTS

### 4.1 EXPERIMENTAL SETUPS

#### 4.1.1 DATASETS.

We conduct extensive experiments on three real-world datasets: Neural 3D Video, Technicolor, and Nvidia Dynamic Scenes. Training is performed on two or three selected views that adequately cover the scene content, and evaluation is carried out on **all the remaining views**.

**Neural 3D Video Dataset.** (Li et al., 2022) It contains six indoor multi-view video sequences, each captured by 18-21 synchronized cameras at a resolution of 2704×2028 and 30 fps. Following Yang et al. (2023), we perform both training and evaluation on downsampled videos by a factor of two, using 300 frames per scene.

**Technicolor Dataset.** (Sabater et al., 2017) It consists of five indoor multi-video video sequences, each captured by a 4×4 synchronized camera array at a resolution of 2048×1088. Following Attal et al. (2023), we conduct both training and evaluation at full resolution, using 50 frames per scene.

**Nvidia Dynamic Scenes Dataset.** (Yoon et al., 2020) This dataset contains of six outdoor multi-view video sequences, each captured by 12 synchronized cameras at 1920×1080 and 60Hz. We use half-resolution frames and 100 frames per scene for training and evaluation.

#### 4.1.2 IMPLEMENTATION DETAILS.

Our implementation is based on Pytorch (Paszke et al., 2019) framework. We train each scene for 30000 iterations. In each training iteration, we sample one input view and one generated view. The first 3000 iterations are trained with a vanilla setting as a warm-up stage, followed by training with both camera optimization and the distortion field, where camera optimization is stopped after 7000 iterations. For generated views, we employ a camera-controlled image-to-video diffusion model Yu et al. (2024) which generates $L = 25$ frames per sequence. For uncalibrated generated views and cameras, we obtain coarse pose and point cloud initialization using COLMAP (Schönberger & Frahm, 2016; Schönberger et al., 2016) at $t = 0$. The weighting factors $\lambda$, $\lambda_1$, $\lambda_2$, $\lambda_p$ and $\lambda_s$ are set to 0.2, 0.02, 0.2, 0.1 and $10^{-4}$, respectively. All experiments are conducted on an Nvidia H800 GPU to ensure fair comparisons.

**Aligning camera poses on test views.** In standard 4DGS reconstruction process, the exact camera poses of the test views are usually available, as they can be estimated together with the training views under a shared coordinate system before reconstruction. However, in our experimental setting, the poses of test views are unknown and pose optimization is applied during training. Therefore, aligning test-view poses before rendering is essential. Following Fan et al. (2024), we freeze the trained

Table 1: **Qualitative comparisons on Technicolor (Sabater et al., 2017), Neural 3D Video (Li et al., 2022), and Nvidia Dynamic Scenes (Yoon et al., 2020) Datasets.** The first and second best performances are highlighted in red and yellow. Our method shows superior performance compared to all baseline methods across all metrics. Note that MonoFusion* is our reproduced version.

| Method | Technicolor | | | Neural 3D Video | | | Nvidia Dynamic Scenes | | |
|---|---|---|---|---|---|---|---|---|---|
| | PSNR↑ | SSIM↑ | LPIPS↓ | PSNR↑ | SSIM↑ | LPIPS↓ | PSNR↑ | SSIM↑ | LPIPS↓ |
| HyperReel (Attal et al., 2023) | 14.14 | 0.453 | 0.616 | 15.63 | 0.582 | 0.500 | 19.88 | 0.528 | 0.396 |
| 4DGaussians (Wu et al., 2024a) | 16.20 | 0.505 | 0.552 | 17.40 | 0.673 | 0.320 | 16.81 | 0.372 | 0.516 |
| 4D-Rotor (Duan et al., 2024) | 14.85 | 0.426 | 0.581 | 18.20 | 0.708 | 0.357 | 19.38 | 0.508 | 0.389 |
| RealTime4DGS (Yang et al., 2023) | 16.53 | 0.510 | 0.542 | 17.31 | 0.649 | 0.442 | 17.91 | 0.479 | 0.426 |
| MonoFusion* (Wang et al., 2025b) | 17.97 | 0.578 | 0.352 | 18.43 | 0.738 | 0.270 | 20.22 | 0.590 | 0.192 |
| Ours | 23.15 | 0.728 | 0.299 | 21.91 | 0.789 | 0.258 | 24.81 | 0.794 | 0.150 |

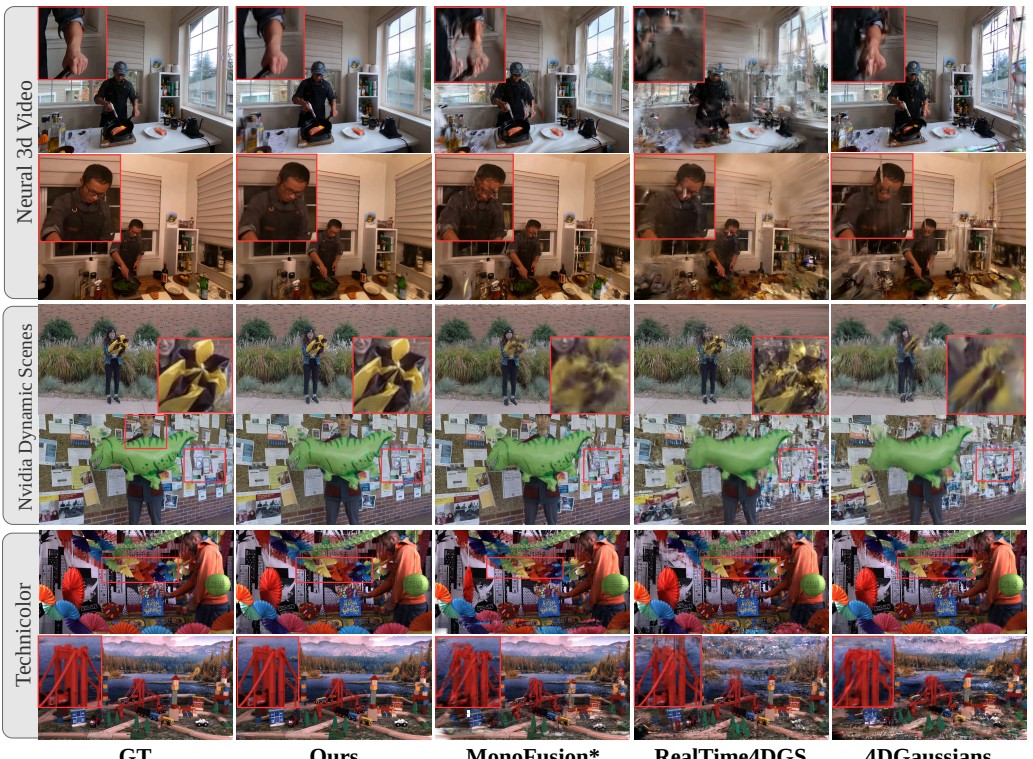

| GT | Ours | MonoFusion* | RealTime4DGS | 4DGaussians |

Figure 4: **Qualitative Comparisons of different methods on Technicolor (Sabater et al., 2017), Neural 3D Video (Li et al., 2022), and Nvidia Dynamic Scenes (Yoon et al., 2020) Datasets.** We conduct comparisons with representative dynamic scene reconstruction methods: MonoFusion (Wang et al., 2025b), 4DGS (Wu et al., 2024a), 4D-Rotor (Duan et al., 2024), and Realtime4DGS (Yang et al., 2023). MonoFusion* is our reproduced version. Our method significantly outperforms other baselines, producing visually reliable results with sharper details. Please zoom in for more details. For more qualitative results, please refer to the supplementary material.

4DGS parameters and optimize only the camera poses of test views. This optimization minimizes the $l_1$ photometric error between rendered and ground-truth images, yielding more accurate alignment of the rendered results with test views. Such alignment eliminates errors caused by inaccurate poses, ensuring a fairer comparison.

## 4.2 COMPARISONS

We adopt PSNR, SSIM (Wang et al., 2004), and LPIPS (Zhang et al., 2018) as evaluation metrics for comparing the rendering quality of our method against baselines. As the compared baselines do not include pose optimization, we adopt the ground-truth poses for both training and rendering. While this setup provides the baselines with a mild advantage, it does not overcome the fundamental chal-

Table 2: **Ablation studies.** We random select one representative scene from Technicolor (Sabater et al., 2017) and Nvidia Dynamic Scenes (Yoon et al., 2020) Datasets to ablate our framework.

| Setting | Balloon1 | | | Train | | | Average | | |
|---|---|---|---|---|---|---|---|---|---|
| | PSNR↑ | SSIM↑ | LPIPS↓ | PSNR↑ | SSIM↑ | LPIPS↓ | PSNR↑ | SSIM↑ | LPIPS↓ |
| w/o distortion field | 23.76 | 0.715 | 0.259 | 17.09 | 0.426 | 0.608 | 20.42 | 0.570 | 0.434 |
| w/o time axis | 24.94 | 0.798 | 0.156 | 17.25 | 0.480 | 0.458 | 21.09 | 0.639 | 0.307 |
| w/o pose axis | 24.81 | 0.793 | 0.158 | 17.38 | 0.462 | 0.469 | 21.09 | 0.627 | 0.314 |
| w/o pose optimization | 24.92 | 0.804 | 0.133 | 18.96 | 0.569 | 0.336 | 21.94 | 0.686 | 0.235 |
| Ours | **25.31** | **0.810** | **0.127** | **21.56** | **0.656** | **0.264** | **23.44** | **0.733** | **0.195** |

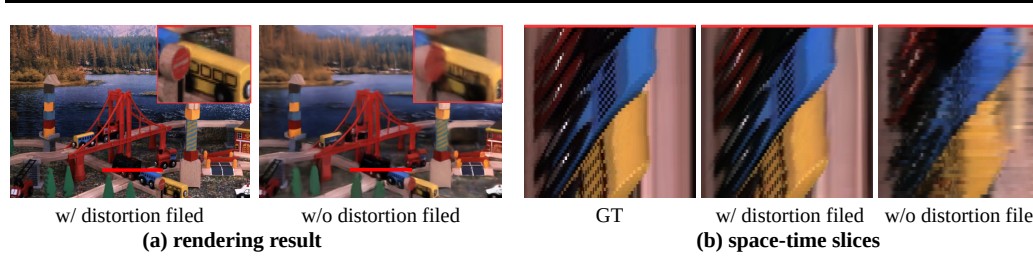

| w/ distortion filed | w/o distortion filed | | GT | w/ distortion filed | w/o distortion filed |
|---|---|---|---|---|---|
| **(a) rendering result** | | | | **(b) space-time slices** | |

Figure 5: **Qualitative ablation results.** Rendering results (a) and space-time slices (b), constructed by concatenating the red pixel locations across all time steps, demonstrate that direct reconstruction from diffusion observations results in severe blur and temporal instability.

lenge caused by sparse camera inputs. Qualitative and quantitative results are shown in Fig. 4 and Tab. 1, respectively. The visualization results and metrics demonstrate that our approach consistently produces sharper details, more stable dynamics, and fewer artifacts under sparse-camera settings, achieving substantially better visual quality and spatio-temporal consistency than all baseline methods. For general 4DGS methods such as 4DGaussians (Wu et al., 2024a) and RealTime4DGS (Yang et al., 2023), their performance drops significantly due to the ill-posed nature of sparse-camera condition, producing broken geometry, noisy renderings and missing details in dynamic regions. These results highlight their reliance on dense and well-aligned inputs. Compared with general 4DGS, MonoFusion (Wang et al., 2025b) benefits from various geometric priors and thus offers relatively high-quality initialization. This leads to noticeable improvements in static background regions. However, it still produces artifacts and geometric misalignments, especially in complex dynamic regions. For instance, in the Nvidia Dynamic Scenes dataset, rapid motions and occlusions cause its priors to fail to capture fine-grained dynamics, resulting in unreliable constraints and degraded reconstructions. The limitation mainly stems from the insufficient quality and robustness of the imposed priors under highly dynamic settings. In contrast, Our method leverages generative priors and explicitly disentangles distortions using the proposed spatio-temporal distortion field, achieving spatio-temporal consistency even under challenging conditions, including large view ranges and complex foreground dynamics. For example, in the Neural 3D Video dataset, our method reconstructs both static and dynamic regions with fine-grained details, while in the Nvidia Dynamic Scenes dataset, it preserves temporal stability despite outdoor motions. These results demonstrate that our method not only mitigates artifacts and geometric failures observed in baselines but also provides robust and reliable reconstructions across diverse datasets.

## 4.3 ABLATION STUDIES

We randomly select three scenes, each from different datasets—Technicolor (Sabater et al., 2017), Neural 3D Video (Li et al., 2022), and Nvidia Dynamic Scenes (Yoon et al., 2020), to ablate our framework. The analysis primarily evaluates the effectiveness of the proposed Spatio-Temporal Distortion Field. In addition, we examine the impact of camera pose optimization.

**Spatio-Temporal Distortion Field.** The '*w/o distortion field*' variant removes the proposed Spatio-Temporal Distortion Field and directly reconstructs 4D scenes with generated images. As shown in Tab. 2 and Fig. 5(a), it produces severe blur result due to the spatio-temporal inconsistencies introduced by generative observations, whereas our distortion field substantially mitigates such problem and significantly improves rendering quality. The space-time slice depicted in Fig. 5(b) is obtained

Table 3: **Ablation study with alternative video diffusion models (VDMs) on Cook Spinach.**

| Cook Spinach | PSNR↑ | SSIM↑ | LPIPS↓ |
|---|---|---|---|
| ViewCrafter (w/o STDF) | 21.42 | 0.775 | 0.302 |
| ViewCrafter (w STDF) | 23.93 | 0.832 | 0.232 |
| ReCamMaster (w/o STDF) | 21.97 | 0.756 | 0.315 |
| ReCamMaster (w STDF) | 23.61 | 0.806 | 0.247 |

Figure 6: **Qualitative ablation results.**

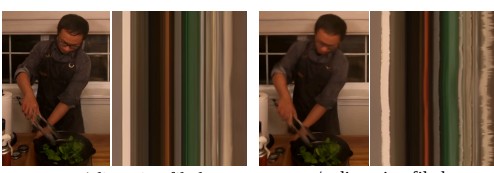

w/ distortion filed          w/o distortion filed

by concatenating the red pixels on Fig. 5(a) across all time steps. Compared with reconstructing directly with generated images, our method yields better temporal consistency and closer alignment with the ground truth motion. These results demonstrate the effectiveness of the proposed distortion-aware 4DGS design.

In addition, to further validate the effectiveness of STDF for unified modeling of spatial and temporal inconsistencies, we conducted the following two experiments. The '*w/o temporal index*' and '*w/o pose index*' variants remove the $t$-axis and $s$-axis in the STDF respectively, thereby modeling inconsistencies only along spatial or temporal dimensions. As shown in Tab. 2, the noticeable performance degradation indicates that generative inconsistencies manifest across both space and time. This validates the necessity of our two-dimensional temporal design, which more effectively aligns spatio-temporal content and improves rendering quality.

**Pose Optimization.** In the *w/o pose optimization* variant, camera poses are fixed during training without refinement. As reported in Tab. 2, incorporating pose optimization substantially enhances reconstruction quality. This demonstrates that generative distortions can severely bias pose estimation, and correcting them during training is essential for forming consistent 4D reconstructions.

**Ablation with Alternative VDM** As discussed earlier, inconsistency in generated frames are inherent to existing VDMs. To further validate the applicability of our approach, we conduct experiment on another state-of-the-art camera-controlled VDM, *i.e.*, ReCamMaster (Bai et al., 2025). As shown in Tab. 3, neither ViewCrafter nor ReCamMaster is able to directly reconstruct photorealistic 4D scenes, highlighting that inconsistencies in VDM-generated frames severely hinder the convergence of 4D content. In contrast, our method achieves significant improvements. When using ViewCrafter as the generative prior, our pipeline yields an increase of 2.51db in PSNR, and when using ReCam-Master as the prior, a similar gain of 1.76db in PSNR is observed. These results demonstrate the capability of our approach to ensure spatio-temporal consistency even when conditioned on different generative models.

## 5 DISCUSSION AND CONCLUSION

**Limitations.** While our method effectively models spatio-temporal inconsistencies, generative models may still produce large-scale errors in complex or ambiguous scenes, such as structural collapse or content hallucination, which cannot be rectified by 4D primitive deformations and can severely impair 4D reconstruction. A promising future direction is to leverage reliably reconstructed content to correct generative outputs by combining 3D priors with the generative capabilities of 2D diffusion models, leading to more robust and coherent dynamic reconstructions.

**Conclusion.** We propose a novel framework that leverages generative models for dynamic scene reconstruction from sparse camera inputs. Our core insight is that spatio-temporal inconsistencies in generative observations are the primary obstacle to achieving high-quality dynamic reconstruction. To address this, we introduce the Spatio-Temporal Distortion Field to model these inconsistencies across both spatial and temporal dimensions, which is then incorporated into a unified framework that jointly optimizes pose, rendering, and smoothness constraints for stable convergence. Our experiments and ablations demonstrate the effectiveness of our method and the critical role of unified spatio-temporal modeling.

To the best of our knowledge, this is the first sparse-camera 4D reconstruction method comprehensively evaluated on standard multi-camera dynamic scene benchmarks, paving the way for more practical and accessible immersive 4D scene reconstruction.

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
