# OpenReview forum: "Towards Spatio-Temporally Consistent 4D Reconstruction from Sparse Cameras"
_ICLR.cc/2026/Conference — ICLR 2026 Conference Withdrawn Submission_

### Official Review · Reviewer_z9B6 · 2025-10-29

**Soundness:** 3
**Presentation:** 3
**Contribution:** 3
**Rating:** 6
**Confidence:** 3

**Summary:**

This paper presents a sparse-camera 4D reconstruction framework that leverages generative observations to achieve high-quality dynamic scene rendering without relying on dense camera setups. The core component, the SpatioTemporal Distortion Field (STDF), models inconsistencies across spatial and temporal dimensions to produce spatio-temporally consistent and photorealistic reconstructions. While the method demonstrates strong empirical performance, the paper lacks sufficient theoretical justification or high-level analysis explaining why the STDF effectively mitigates inconsistencies in the pseudo-labels.

**Strengths:**

1.The proposed method delivers strong performance, achieving noticeable improvements over existing approaches.

2.The motivation is meaningful, as the authors claim this to be the first work enabling sparse-camera 4D reconstruction on dynamic scene benchmarks. However, this extension appears somewhat natural, given that similar advancements in supporting sparse-view inputs have already been demonstrated in 3D Gaussian Splatting (3DGS) and NeRF.

**Weaknesses:**

1. The paper lacks sufficient theoretical justification or high-level analysis to explain why the proposed STDF effectively mitigates inconsistencies in the pseudo-labels. I hope the authors can provide deeper theoretical insights or analytical evidence to substantiate their claims, rather than relying solely on empirical performance.

**Questions:**

1.Lack of Theoretical Justification:
The paper lacks sufficient theoretical justification or in-depth analysis to explain why the proposed SpatioTemporal Distortion Field (STDF) effectively mitigates inconsistencies in the pseudo-labels. More theoretical or high-level analysis is expected to strengthen the validity of the method beyond empirical evidence.
2.Comparison with Existing Methods:
The proposed STDF appears to be conceptually related to HexPlane [1]. A clearer comparison with HexPlane should be included to highlight the novelty and distinct contributions of the proposed approach.
[1] Cao A, Johnson J. Hexplane: A fast representation for dynamic scenes[C]//Proceedings of the IEEE/CVF Conference on Computer Vision and Pattern Recognition. 2023: 130-141.
3. Clarification on View Treatment
It is unclear whether the generated views and ground-truth views are treated identically throughout the pipeline or if there are additional differences beyond the loss supervision. The authors should clarify whether any distinct processing, weighting, or architectural handling is applied to these two types of views during training.
4. Evaluation under Different Sparsity Levels
Although the paper is claimed under sparse view condition. I think the author should give more dense view results to show the performance of the proposed method in different level sparse view, e.g., 1,2,4,8 gt view.etc.

---

### Official Review · Reviewer_kitB · 2025-10-30

**Soundness:** 3
**Presentation:** 3
**Contribution:** 2
**Rating:** 6
**Confidence:** 3

**Summary:**

This paper presents a framework for sparse-camera dynamic scene reconstruction using diffusion-based generative observations. The main contribution is the Spatio-Temporal Distortion Field (STDF), which models geometric and temporal inconsistencies in a 5D latent space. By coupling this field with pose refinement and multi-term loss functions, the method achieves spatio-temporal consistency under limited camera views. Experiments on Neural 3D Video, Technicolor, and NVIDIA Dynamic Scenes show performance improvements over baselines such as MonoFusion, RealTime4DGS, and 4D-Rotor, with better temporal coherence and fewer artifacts.

**Strengths:**

1.The problem is well-motivated. Sparse-camera dynamic reconstruction is both challenging and practically important for many field.

2.Evaluation across three large-scale benchmarks, with quantitative and visual superiority, demonstrates robustness.

3.The paper systematically isolates STDF components (time/pose axes, pose optimization) and shows consistent performance gains.

4..Discarding the distortion field after training ensures zero runtime overhead, which is an elegant design choice.

**Weaknesses:**

1.Limited conceptual novelty: The overall framework remains a structured combination of known components (K-Planes, 4DGS, and VDM priors). STDF acts as a deformation correction layer rather than introducing a new representation or optimization principle.

2.The paper does not deeply justify how the STDF mathematically stabilizes generative inconsistencies or why its 9-plane decomposition is optimal. There are few visualizations or theoretical insights into its behavior.

3.Performance relies heavily on the fidelity of diffusion-generated frames. Although Table 3 tests ViewCrafter and ReCamMaster, it remains unclear how robust the method is under more severe generative noise.

4.While impressive results are achieved, all evaluations appear to be interpolation-based. It remains unclear whether the method generalizes to unseen or extrapolated viewpoints.

5.The paper reports quantitative comparisons with ground-truth frames, but does not clarify how diffusion-generated inputs are stabilized. Since diffusion priors can introduce random lighting or texture variations, reproducibility and color consistency across runs are unclear.

**Questions:**

The authors could better illustrate how STDF corrects distortions across time and space compared to simpler 4D regularization. Are there visualizations showing the learned distortion fields? Additionally, could similar spatio-temporal stability be achieved through flow or pose based constraints without diffusion priors?
Furthermore, the paper does not specify which camera views are used for training and testing, nor how the 2, 4, and 6 views settings are distributed. Providing camera index references or a schematic of view layouts would help clarify whether evaluations are conducted on interpolated or extrapolated views, and ensure reproducibility and fair comparison across baselines.

---

### Official Review · Reviewer_SSnY · 2025-10-31

**Soundness:** 2
**Presentation:** 3
**Contribution:** 3
**Rating:** 4
**Confidence:** 3

**Summary:**

This paper addresses the challenge of 4D dynamic scene reconstruction from sparse camera viewpoints. The authors propose a Spatio-Temporal Distortion Field (STDF) designed to model and compensate for spatio-temporal inconsistencies in frames generated by video diffusion models. The method leverages generative priors during training by learning distortion patterns, while reportedly introducing no additional computational overhead during inference. Experimental results on multiple benchmark datasets indicate improved performance over baseline methods in quantitative metrics.

**Strengths:**

(1)	Originality: The primary strength of this paper lies in its novel and highly insightful core idea of the Spatio-Temporal Distortion Field (STDF), which provides an elegant and direct solution to the fundamental problem of inconsistencies when using generative models for 4D reconstruction.

(2)	Clarity: The method is clearly presented, supported by lucid tables and visualizations that effectively communicate the method's advantages and results. Furthermore, the paper is logically structured with adequate visualizations that support understanding of the key concepts.

(3)	Quality: The experimental evaluation covers relevant benchmarks and baseline comparisons, providing evidence for the method's effectiveness.

(4)	Significance: The work addresses the practical problem of sparse-view 4D reconstruction, which is of importance to the field.

**Weaknesses:**

The paper presents a strong and novel method, but the following points would benefit from further clarification and analysis to fully substantiate the claims and provide a more comprehensive evaluation.

(1)	Lack of Failure Case Analysis: The limitation regarding large-scale generative errors (e.g., structural collapse or content hallucination) is appropriately acknowledged in Section 5, "Limitations". However, providing visual examples of these failure cases would offer a more comprehensive and tangible understanding of the method's current boundaries and robustness.

(2)	Uncertainty in View Selection Protocol: As stated in Section 4.1.1, Line 295, training is performed on "two or three selected views that adequately cover the scene content." The specific criteria for this selection are not detailed. An analysis of the performance sensitivity to different sparse view choices would strengthen the claim of the method's practical robustness.

(3)	Incomplete Evaluation of Core Claims: The central claim of the work is achieving superior "spatio-temporally consistent" reconstruction, as emphasized in Section 4.2, Line 403. However, the experimental validation in Table 1 relies primarily on per-frame image quality metrics. A more direct validation of spatio-temporal consistency would be strengthened by incorporating metrics for temporal stability (e.g., t-LPIPS) and geometric quality, the latter being a focus of the MonoFusion baseline[1].

(4)	Unquantified Computational Overhead: While the paper correctly notes that the STDF introduces zero overhead during inference, the computational cost during training is not reported. A comparison of training time or resource usage relative to the main baselines would help practitioners fully assess the efficiency trade-offs of the proposed framework.

[1]	Wang, Zihan, et al. "MonoFusion: Sparse-View 4D Reconstruction via Monocular Fusion." Proceedings of the IEEE/CVF International Conference on Computer Vision. 2025.

**Questions:**

(1)	To better understand the practical limitations discussed in Section 5, could you provide visual examples of typical failure cases caused by major generative errors?

(2)	Could you please elaborate on the protocol used for selecting the 2-3 training views mentioned in Line 295? Additionally, have you conducted any sensitivity analysis to evaluate how the performance varies with different view selections? This would be very valuable for assessing the method's robustness in practical scenarios.

(3)	For weakness – (3), given the paper's focus on spatio-temporal consistency, I would suggest that the evaluation could be even more compelling. Would it be possible to provide additional quantitative results using metrics specifically for temporal stability (e.g., t-LPIPS) and for geometric quality (e.g., depth accuracy like AbsRel, following the evaluation in MonoFusion[1])? This would offer direct evidence for the improvements in these specific dimensions.

(4)	Could you provide more details on the computational cost of training? For instance, a comparison of the total training time per scene or GPU memory usage against your main baselines.

---

### Official Review · Reviewer_qHpx · 2025-11-01

**Soundness:** 3
**Presentation:** 3
**Contribution:** 2
**Rating:** 4
**Confidence:** 4

**Summary:**

This work proposes a framework for 4D reconstruction from sparse camera inputs by leveraging diffusion-generated images as auxiliary observations. The core idea is the Spatio-Temporal Distortion Field (STDF), which models inconsistencies in generative observations across both spatial and temporal dimensions. The framework jointly optimizes pose, rendering, and smoothness terms to achieve spatio-temporal consistency. Experiments on standard 4D reconstruction benchmarks show improvements over existing baselines.

**Strengths:**

* The idea of leveraging diffusion-generated outputs to improve 4DGS reconstruction and rendering is well-motivated and sound.


* The code is provided, which facilitates reproduction and verification.


* The manuscript is well-structured and easy to follow.

**Weaknesses:**

* **Inconsistent or misleading gains over the baseline model.** As reported in the ablation study (Tab. 2), the improvement over the baseline (“w/o distortion field”) is about 2 dB in PSNR. However, when compared to the same baseline (“4DGaussians”) in the main results (Tab. 1), the improvement rises to around 7 dB. If the major difference comes from using diffusion-generated images as input, it would be more appropriate to use these generated images for all comparison models. Otherwise, the large improvement shown in Tab. 1 could be misleading.


* **Lack of comparison with more related work.** It would be beneficial to include comparisons with other recent methods that also utilize diffusion models to enhance 4DGS, such as GS-GS [ref 1]. Since GS-GS also reports significantly higher PSNR (27.13) on Neural 3D Video, adopting its settings for direct comparison would provide a fairer assessment of the proposed method’s improvement.


* **Unclear explanation of limitations.** The paper states that the method is not effective in “complex or ambiguous scenes,” yet the current experiments already use real-world captured data, which appear complex. Providing concrete examples and in-depth analysis would help clarify the actual failure cases and better illustrate the trade-offs of the proposed approach.

### References:

[ref 1] Kong, Hanyang, Xingyi Yang, and Xinchao Wang. "Generative Sparse-View Gaussian Splatting." CVPR 2025.

**Questions:**

What prevents this method from being applied to monocular videos? Since the related work Shape-of-Motion achieves 4D reconstruction from a single video, could the proposed framework be extended to monocular video settings?

---

### Note · Authors · 2025-11-14

I have read and agree with the venue's withdrawal policy on behalf of myself and my co-authors.